# Domoic Acid and *Pseudo-nitzschia* spp. Connected to Coastal Upwelling along Coastal Inhambane Province, Mozambique: A New Area of Concern

**DOI:** 10.3390/toxins13120903

**Published:** 2021-12-15

**Authors:** Holly Kelchner, Katie E. Reeve-Arnold, Kathryn M. Schreiner, Sibel Bargu, Kim G. Roques, Reagan M. Errera

**Affiliations:** 1School of Renewable Natural Resources, Louisiana State University and Agricultural and Mechanical College, Baton Rouge, LA 70803, USA; hkelch@umich.edu; 2Cooperative Institute for Great Lakes Research, University of Michigan, Ann Arbor, MI 48108, USA; 3All Out Africa Marine Research Centre, Praia do Tofo, Inhambane 1300, Mozambique; katie.reevearnold@gmail.com (K.E.R.-A.); kim@alloutafrica.com (K.G.R.); 4Large Lakes Observatory, University of Minnesota Duluth, Duluth, MI 55812, USA; kschrein@d.umn.edu; 5Department of Chemistry & Biochemistry, University of Minnesota Duluth, Duluth, MN 55812, USA; 6Department of Oceanography and Coastal Sciences, College of Coast and Environment, Louisiana State University and Agricultural and Mechanical College, Baton Rouge, LA 70803, USA; sbargu@lsu.edu; 7National Oceanic and Atmospheric Administration Great Lakes Environmental Research Laboratory, Ann Arbor, MI 48108, USA

**Keywords:** phycotoxin, monitoring, biodiversity hotspot, CHEMTAX, Western Boundary System, *Trichodesmium*, upwelling index

## Abstract

Harmful algal blooms (HABs) are increasing globally in frequency, persistence, and geographic extent, posing a threat to ecosystem and human health. To date, no occurrences of marine phycotoxins have been recorded in Mozambique, which may be due to absence of a monitoring program and general awareness of potential threats. This study is the first documentation of neurotoxin, domoic acid (DA), produced by the diatom *Pseudo-nitzschia* along the east coast of Africa. Coastal Inhambane Province is a biodiversity hotspot where year-round *Rhincodon typus* (whale shark) sightings are among the highest globally and support an emerging ecotourism industry. Links between primary productivity and biodiversity in this area have not previously been considered or reported. During a pilot study, from January 2017 to April 2018, DA was identified year-round, peaking during Austral winter. During an intense study between May and August 2018, our research focused on identifying environmental factors influencing coastal productivity and DA concentration. Phytoplankton assemblage was diatom-dominated, with high abundances of *Pseudo-nitzschia* spp. Data suggest the system was influenced by nutrient pulses resulting from coastal upwelling. Continued and comprehensive monitoring along southern Mozambique would provide critical information to assess ecosystem and human health threats from marine toxins under challenges posed by global change.

## 1. Introduction

Wind-driven upwelling is a key driver of physical, biogeochemical, and ecological variability within coastal ocean environments, as nutrients delivered by coastal upwelling stimulate growth of phytoplankton, ultimately fueling diverse and productive marine ecosystems. The dynamics of wind-driven upwelling are fundamentally driven by large-scale atmospheric processes and have been extensively studied along the Eastern Boundary System (EBS) such as the Canary, California, Benguela, and Humbolt regions [1]. The Western Boundary System (WBS) such as the Gulf Stream, Kuroshio, and Agulhas regions are generally oligotrophic compared to EBS [2], and are known for mesoscale eddy activity and coastal upwelling [3,4,5]. Mesoscale eddies that upwell nutrients from deeper waters play an important role, increasing diversity within the food web due to the pulsed effect of introducing nutrients to the system [3,6,7].

Harmful algal blooms (HABs) are a natural phenomenon, which are increasing globally in frequency, persistence, and geographic extent [8,9,10,11,12]. These blooms can cause substantial harm to coastal societies through health impacts, increased costs associated with bloom management, and decreased revenue during fisheries and beach closures [13,14,15]. Domoic acid (DA), a toxin produced by some *Pseudo-nitzschia* spp., has been widely monitored since the first recorded outbreak resulted in the death of at least three people following the consumption of contaminated shellfish [16]. Domoic acid is a potent neurotoxin produced by several species of *Pseudo-nitzschia*, leading to both acute and chronic toxicity in mammals, fish, and birds [17,18,19,20,21]. Human consumption of highly contaminated shellfish can lead to Amnesic Shellfish Poisoning (ASP); symptoms include gastrointestinal distress, confusion, disorientation, seizures, short-term memory loss, and, in rare cases, death [16,22,23]. No human fatalities have been reported since the widespread implementation of monitoring programs [24,25,26].

*Pseudo-nitzschia* spp. is a cosmopolitan diatom, associated with upwelling regions, particularly the EBS, including the California, Humboldt, Canary, and Benguela currents [27,28], due to its ability to take advantage of disturbance or transitional ecological states. Blooms of *Pseudo-nitzschia* in EBS typically occur between late spring and summer when seasonal upwelling introduces key nutrients, particularly nitrogen [29,30,31]. Blooms of *Pseudo-nitzschia* along the WBS typically occur in shallow bays or estuaries where nutrients are terrestrially sourced, such as along the coast of China and eastern United States [32,33,34]. Incidences of DA have been less prevalent within the WBS than the EBS [24]. While the EBS has received attention for an increase in *Pseudo-nitzschia* abundance and DA events, coastal zones worldwide are being impacted by DA due to increased frequency of warm ocean anomalies [35], and the WBS could be a potential area of concern for future toxic blooms. In the southern hemisphere, DA emerged as a potential threat to fisheries within the WBS in New Zealand in the 1990s [36,37]. Along the east coast of South America, a bloom of *Pseudo-nitzschia australis* produced detectable levels of DA within the Argentine Sea in 2000 [38,39]. 

HABs within the Southwestern Indian Ocean (SWIO) are understudied due to a lack of monitoring infrastructure, despite the presence of potentially harmful species and documented marine toxins within the region [40]. Okadaic acid and ciguatoxin, produced by dinoflagellates, have been recorded along the east coast of Madagascar and nearby islands including Mauritius, the French Islands, Comoros, and Seychelles [40,41,42]. The consumption of a single shark contaminated with ciguatoxin along the coast of Madagascar led to the hospitalization of 188 people and the death of 50 people in 1993 [41]. South Africa has the most developed HAB monitoring program within the SWIO and has reported multiple derivatives of okadaic acid and yessotoxins within mussels along the western side of the country [43]. While *Pseudo-nitzschia multiseries* has been responsible for DA accumulation in mussels within Algoa Bay, South Africa [44], no marine toxins have caused shellfish closures along the eastern coast [45]. To date, there have been no reported HAB-related events in Mozambique [40], though *Pseudo-nitzschia* is among the coastal diatom assemblage [46].

The plankton communities of Delagoa Bight in Maputo Bay and Sofala Bank along Mozambique’s coastline [46,47] have high primary productivity that support higher trophic level diversity along southern Mozambique [48,49,50]. Among Mozambique’s coastal regions, Inhambane Province, located between Delagoa Bight and Sofala Bank, is known as a biodiversity hotspot [51,52], with whale sharks, manta rays, migrating humpback whales, bottlenose dolphins, humpback dolphins, dugongs, sea turtles, small-eye stingray, guitar sharks, corals, and many reef fishes utilizing the region [51,52,53,54,55,56]. This biodiversity supports a prominent tourism industry for SCUBA diving, snorkeling, surfing, and fishing [57]. In addition, roughly two-thirds of Mozambique’s population lives along the coast and relies on marine resources as a source of food and income [58,59].

Currently, there are limited studies contributing to the understanding of phytoplankton dynamics in coastal Mozambique [46,60,61,62,63]. Here we present the first known examination of the phytoplankton community composition within the coastal system of Inhambane Province and document the occurrence of DA from January 2017 to August 2018. We examine plankton abundance in relationship to inorganic dissolved nutrients (nitrogen (N) phosphorus (P) and silica (Si)) and other environmental parameters at four sites along the coast of Inhambane Province. We intend for this research to serve as a baseline to identify future changes to the coastal area, provide essential baseline data of the phytoplankton assemblage with the region, and initiate local monitoring of DA and other potential phycotoxins in Inhambane Province, Mozambique, to help ensure human and ecosystem health.

## 2. Results

### 2.1. Sites

During pilot DA monitoring from January 2017 to April 2018, all samples were collected at the soft substrate region. Unless otherwise noted, results refer to the intense study that occurred between May and August 2018. No significant difference in temperature (T; *p* = 0.81), dissolved oxygen (DO; *p* = 0.76), or salinity (*p* = 0.75) was observed between sites. Dissolved inorganic nutrient concentrations and particulate DA had no significant differences in concentrations among sites (N: *p* = 0.573, P: *p* = 0.300, Si: *p* = 0.479), which suggest that the coastal system from coastal Barra Beach to Praia de Jangamo (coastal Inhambane Province) is well-mixed and can be considered as a single region (Figure 1). Data from the four sites were combined into one region.

### 2.2. Upwelling Index

Localized upwelling index was used to estimate upwelling transport flux (m^3^ s^−1^ km^−1^) along the coast of Inhambane Province (Figure 2). Fluctuations lasting a few days to a few weeks occurred through May into June 2018. Prolonged increased upwelling flux occurred following the 14 July 2018 and peaked at the end of August (following the study period). This event (approximately the 14th of July through August) hereafter is referred to as a major upwelling event (MUE).

### 2.3. Environmental Conditions

Subsurface T (22.7–25.6 °C) and DO (6.36–7.04 mg·L^−1^) decreased throughout the study period from May through August, while salinity remained relatively consistent (36.58 ± 0.4). A decrease in T was observed after the 14th of July 2018, which corresponded to MUE (Figure 2). Inorganic dissolved Si concentrations ranged from 0.99 mg·L^−1^ to 2.27 mg·L^−1^ with an average value of 1.41 ± 0.39 mg·L^−1^. The MUE had no apparent effect on Si concentration (*p* = 0.82). Dissolved iron concentrations were below detection limits (0.007 mg·L^−1^). Dissolved inorganic N within the system was dominated by NH_4_^+^ with no NO_2_ detected. Total dissolved N concentrations fluctuated within each region and ranged between 1.60 and 4.15 µM·L^−1^ (Figure 3A). Though not statistically significant, the soft substrate site consistently had lower N concentrations (mean 2.15 ± 0.62 µM·L^−1^) than the reef sites (mean 2.66 ± 0.78 µM·L^−1^). Dissolved inorganic P concentration ranged from 0.161 µM·L^−1^ to 0.36 µM·L^−1^ throughout the study period with an average concentration of 0.21 ± 0.04 µM·L^−1^ (Figure 3B). The soft substrate site tended to have a higher concentration of P with a mean of 0.25 ± 0.07 µM·L^−1^, while reef sites had a collectively similar mean of 0.196 ± 0.02 µM·L^−1^. Prior to the upwelling event, the mean N:P ratio was 13.4 ± 3.92, and following the MUE, this ratio decreased to 10.48 ± 2.44 with the exclusion of two samples from the central site, which were high in nitrogen (Figure 3C).

### 2.4. Particulate Organic Matter

The ratio of carbon (C) to N (C:N) within the particulate matter fluctuated throughout the study period but increased following the MUE (Figure 4). Prior to the MUE, the small fraction had lower average concentrations of both C (0.0087 ± 0.0047 mg·L^−1^) and N (0.0024 ± 0.0011 mg·L^−1^) than the large fraction (0.0138 ± 0.0050 mgC·L^−1^, 0.0031 ± 0.0011 mgN·L^−1^) (Figure 5A). Regressions between cellular C:N had coefficients of 0.144 (*R*^2^ = 0.395) and 0.196 (*R*^2^ = 0.836) for small and large fractions, respectively. Average C:N ratio was 4.07 ± 1.59 for the small fraction and 4.42 ± 0.61 for the large fraction.

Particulate organic matter (POM) concentrations increased across all regions following the MUE (Figure 5B). The small fraction showed a decreased coefficient of 0.102 (*R*^2^ = 0.881), whereas the large fraction maintained a similar relationship of 0.197 (*R*^2^ = 0.90). Average cellular nutrient concentrations of small (0.0209 ± 0.0120 mgC·L^−1^, 0.0038 ± 0.0013 mgN·L^−1^) and large (0.0224 ± 0.0089 mgC·L^−1^, 0.0046 ± 0.0019 mgN·L^−1^) fractions increased, and average C:N ratios exhibited an increase of 0.24 for the small fraction and a 0.12-fold-change for the large fraction (Figure 5B). The high value measured within the small fraction corresponded to the high chlorophyll *a* seen on the 6th of August 2018 (Figure 6). One sample of the large fraction contained a crustacean larva (~10 mm), likely contributing to the high concentration of C and N.

### 2.5. Phytoplankton Composition

Chlorophyll *a* (chla) concentration was used as a proxy for productivity and was significantly affected by T (*p* < 0.05). Low chla concentrations were observed between the 24th of May through the 20th of July and increased following the MUE event (Figure 6). Concentrations ranged from 4 to 234 ng·L^−1^ prior to the MUE and from 19 to 794 ng·L^−1^ following the MUE.

The phytoplankton community was dominated by diatoms (Figure 7) based on photopigment signatures [64] and subsequent CHEMTAX analysis (Appendix A) [65]. The diatom assemblage was composed of chain-forming genera including *Pseudo-nitzschia* spp., *Bacteriastrum* spp., *Thalassiosira* spp., *Guinardia* spp., and *Chaetoceros* spp., and the non-chain forming *Rhizosolenia* spp. Prevalent dinoflagellates included *Protoperidinium* spp., *Ceratium* spp., and *Pyrophacus* spp. *Alexandrium* spp., *Dinophysis* spp., *Prorocentrum* spp., and *Akashiwo* spp. were also observed in low abundance and have the ability to produce phycotoxins. The primary species identified within the “other” category were *Phaeocystis* spp., a prymnesiophyte, and *Dictyocha* spp., a unicellular flagellate.

*Trichodesmium* spp. tended to be more abundant prior to the MUE (prior to MUE = 4.62 ± 6%, post MUE = 0.66 ± 1.5%), whereas *Pseudo-nitzschia* spp. increased in community proportion across all regions following the MUE. *Pseudo-nitzschia* spp. made up 13.96 ± 15.6% of the community before the MUE, and 40.11 ± 17.5% after the MUE. The soft substrate site had the highest change in *Pseudo-nitzschia* spp. proportion, with a mean 25.2 ± 1.15% before the MUE, and became the dominant species with a mean of 54.5 ± 4.9% following the MUE.

### 2.6. Particulate DA

Surface samples collected as a pilot study between January 2017 and April 2018 combined with samples collected during the intensive sampling period of this study (May–August 2018) revealed the presence of pDA year-round (Figure 8). During pilot sampling, pDA was detected in 96% (*n* = 47) of samples with a range of 0.241 to 49.4 pg pDA·L^−1^. The highest concentration (49.4 pg pDA·L^−1^) was observed on the 11th of June 2017 at the soft substrate region. During the intensive sampling period, there was no statistical difference in pDA between the small fraction and large fraction samples (*p* = 0.62), so values reported are a combined average. During the intensive sampling period, pDA was detected in 80% of samples with a range of 0.295 to 11.71 pg pDA·L^−1^. The highest concentration (11.7 pg pDA·L^−1^) was observed at the soft substrate region on the 27th of July 2018. There was a greater than 2-fold increase of pDA following the MUE. Prior to the MUE, pDA concentrations averaged 0.937 ± 1.25 pg L^−1^ (70% detected, *n* = 20). All samples following the MUE (*n* = 15) had detectable pDA with an average concentration of 4.81 ± 2.41 pg L^−1^. Principle components analysis (PCA; Appendix A) suggested that pDA is negatively influenced by the N:P ratio; pDA is higher when the N:P ratio is low.

## 3. Discussion

Our results indicate that wind-driven upwelling likely reduced N limitation, which allowed for phytoplankton biomass to increase and supported a shift to a *Pseudo-nitzschia* spp.-dominated diatom assemblage. The relative increase in *Pseudo-nitzschia* spp. corresponded with an increase in pDA in the vicinity of Praia do Tofo, Mozambique. To our knowledge, this is the first identification of *Pseudo-nitzschia* spp. and DA in Inhambane Province, Mozambique. Preliminary data suggest a seasonal spike at the onset of Austral winter, with peaks in June of 2017 and July of 2018. Though pDA peaked after a coastal upwelling and decrease in T event, pDA was noted throughout the year. The Mozambique channel is known for having turbid waters due to the formation of eddies within the Mozambique current [66,67], primarily occurring during Austral winter. These mesoscale eddies play an important role in the interannual variability of phytoplankton blooms [63,66] as well as increased biodiversity of higher trophic levels [68]. During our study, satellite imagery identified the occurrence of a cyclonic eddy influencing the coastal waters of Inhambane Province, which corresponded to an increase in the calculated upwelling flux (Appendix A). It is possible that the increase in *Pseudo-nitzschia* spp. and pDA observed in our study were influenced by both wind-driven upwelling and the presence of the mesoscale eddy, and further research is necessary to determine if this trend occurs annually.

On a global scale, DA has been prevalent among EBS and in temperate climate regions where upwelling of nutrients promote toxicity [69,70]. Warming global climate conditions and increased nutrients from terrestrial sources create prime conditions for increases in community dominance by *Pseudo-nitzschia* spp. as well as increased toxicity [26,32,69,71,72,73]. There has been a recent emergence of DA contamination of shellfish fisheries within the WBS including Argentina [18,74], Uruguay [75], Brazil [76], New Zealand [36], and South Africa [44], supporting the increased range of this HAB species. In 2016, the Gulf of Maine recorded its first shellfish fishery closure due to DA concentrations exceeding regulatory limits [24]. *Pseudo-nitzschia* blooms in the Gulf of Maine occur in the fall and persist into the winter, similar to the *Pseudo-nitzschia* bloom and pDA identified in this study near Praia do Tofo, Mozambique. This study is the first documentation of DA along the eastern Africa within a subtropical WBS.

The upwelling event documented here correlated with changes in inorganic nutrient availability and changes to productivity as indicated by increases in chla and seston nutrient composition. Based on global averages of nutrient ratios within marine systems, the Redfield Ratio of 106_C_:16_N_:1_P_:1_Si_:0.01_Fe_ is used as a baseline for determining nutrient limitations [77]. Analysis of both the inorganic dissolved (N:P 13.4 ± 3.92) and seston (C:N 4.8 ± 1.8) nutrient composition suggested the system was slightly N limited prior to the MUE. Silica (Si) is another important nutrient for diatoms [78,79], with averaged ratios of 2:1 (N:Si) and 7:1 (Si:P). The low N:Si ratio further suggests that N is the primary limiting nutrient for diatom growth, and the high Si:P ratio suggests that Si is likely not a limiting nutrient. While low silica conditions have been suggested to provide *Pseudo-nitzschia* spp. an advantage over other diatom species [80], our data indicate high Si available during a transition to a *Pseudo-nitzschia* spp.-dominated phytoplankton assemblage, indicating a different driver was responsible for the shift. While the system experienced a decrease in dissolved nutrient ratio (N:P 10.4 ± 2.44, Figure 3C) following the MUE, we suspect that it was likely due to the rapid uptake of nutrients by phytoplankton as explained by the increase in particulate C and N (Figure 5B). The increase in C:N ratio after the MUE (Figure 4) resulted in a similar C:N ratio to the rest of the southern Indian Ocean (6.5–9.5) [81].

Energetically, high C:N ratios are more efficient at converting energy between trophic levels and therefore are considered more nutritious for primary consumers [82]. Increased seston C:N ratios following the MUE could be a crude indication of higher quality diet; however, *Pseudo-nitzschia* spp. became dominant in response to the MUE, which could have deleterious nutritional consequences on consumers due to decreased diet diversity and the potential for toxin consumption. Fatty acid analysis is suggested as a better indicator of nutritional quality for future research [83], as it measures the essential nutrients often not provided within a monospecific diet [82].

*Pseudo-nitzschia* spp. is well-adapted to shifts in environmental conditions, taking advantage of periotic disturbances such as upwelling events, eddies, or experimental manipulation [30,84]. Following the MUE, our data suggest *Pseudo-nitzschia* spp. was able to take advantage of the periotic disturbance and dominate the community assemblage within the system. Within our study, variability in environmental parameters such as T and nutrients was low between sampling sites; however, some trends require further investigation. Soft substrate sites tended to be low in dissolved N (2.51 ± 0.479 mg·L^−1^) and high in P (0.255 ± 0.074 mg·L^−1^), suggesting N limitation. These sites also consistently had the highest pDA (1.65–11.7 pg pDA L^−1^). PCA analysis suggested a correlation between the low nutrient ratios and *Pseudo-nitzschia* spp. dominance (Appendix A). Another previously explored aspect to this relationship suggests a restriction of DA production under N limited conditions resulting from an insufficient pool of free N for synthesis of the nitrogen-rich toxin [85]. However, a study with multiple species of *Pseudo-nitzschia* revealed various growth responses to nutrient ratios [86], suggesting a more complex and potentially species-specific relationship with nutrients.

Overall, pDA concentrations within our study period were low (0.091–11.71 pg pDA L^−1^) when detected (Figure 8). The solubility of the toxin in water prevents DA from accumulating within ecosystems without a source of production [87,88]; therefore, in areas where DA is known to occur with a seasonal bloom, monitoring is focused during blooms. However, along Inhambane Province, there could be concern for chronic exposure to the ecosystem and possibly human health [89] due to the year-round production of DA, which has been shown to cause declines in cognitive function to the central nervous system [90,91,92]. This study only focused on quantifying pDA within plankton samples, and further investigation into vector organisms is needed to assess the threat of chronic DA exposure.

High concentrations of DA did not correspond to high densities of *Pseudo-nitzschia* spp., corroborating previous studies [24,93,94]. High variance in cell abundance suggests high day-to-day growth fluctuations in the region. Light microscopy identification suggested at least three *Pseudo-nitzschia* spp. present simultaneously along the coast of Inhambane Province, namely *P. multiseries* (Appendix A)*, P. australis* (Appendix A), and *P. turgidula* (Appendix AF,G) [38,44,70], which are all known to produce DA [24]. Prior reports have also identified *P. multiseries*, *P. pungens, P. cuspidate,* and *P. seriata* along the eastern coast of Africa [44,95]. Each individual species of *Pseudo-nitzschia* exhibits specific growth and toxin production dynamics in response to temperature, salinity, irradiance, photoperiod, and variation in nutrients [24], which suggests both cell abundance and toxin concentration must be included in monitoring efforts.

Along the coast of Inhambane Province, the highest toxin concentration was measured in the central, soft substrate region (Figure 1), a common feeding area for aggregations of *Rhincodon typus, Manta alfredi, and M. birostris* [55]. Though not significantly different (*p* = > 0.05), pDA concentrations within the large plankton fraction at the soft substrate region were repeatedly higher than those in the small plankton fraction. Not enough data were collected to support the threat of bioaccumulation; however, there is a pressing need to better understand the risks of toxin bioaccumulation in order to advise local residents on this potential public health issue.

As climate changes, the Praia do Tofo region would benefit from a better understanding of the effects of seasonal influences on plankton assemblage and community dynamics to predict how the system might be affected in the upcoming decades. The current rate of temperature increase in the Indian Ocean will likely have profound effects on surrounding coastal communities [96]. The fluctuation of *Trichodesmium* spp. abundance within the community composition was not significant within the study period but is noted to increase seasonally during the monsoon season in nearby Tanzania [97], promoting the recycling of limited N through nitrogen fixation properties. *Trichodesmium* spp. is also recognized as a harmful species in this area [40] and is predicted to increase in prevalence under elevated surface temperature conditions. Changes to mesoscale processes affecting environmental conditions and nutrient transport are predicted to occur within the Southwestern Indian Ocean [6,98], possibly impacting the factors influencing *Pseudo-nitzschia* spp. assemblage dominance and pDA concentration. Monsoon events, which can affect Mozambique annually from January through March, are predicted to intensify and become more frequent. These events have been shown to increase productivity in the Northern Indian Ocean [99] and south of Madagascar [100]. In addition, increased nitrate during the monsoon season is documented to promote *Pseudo-nitzschia* blooms in Tanzania [101]. Increased rain intensity has also promoted harmful cyanobacteria along coastlines in Tanzania [97]. However, cyclones are a relatively recent threat to southern coastal systems and have not been studied along the coast of Mozambique. Warming is predicted to have delayed effects on the Indian Ocean Dipole, which accounts for interannual variability in rainfall and is likely to decrease total rainfall [102]. Overall, rainfall in southern Africa is expected to become less frequent, but to increase in intensity [103]. These processes are likely to affect coastal systems with decreased terrestrial runoff, such as Inhambane Province, Mozambique.

Anthropogenic nutrient loading can increase growth of harmful algae, including *Pseudo-nitzschia* spp. [28]. Agriculture [104] and aquaculture [59,105] industries are forecasted to continue to grow in Mozambique. As coastal communities develop and increase agriculture and aquaculture production, nutrient levels of freshwater runoff will likely change the affective nutrient input to the coastal phytoplankton communities. Data from this research suggest that the coastal system along Inhambane Province was not highly impacted by freshwater input during the dry season. However, additional research during the wet season (Austral summer) would be necessary to assess if changes in land use will impact this region. Coastal Inhambane Province is a biological hot spot reliant on local fisheries as a food source and economic development through tourism due to the year-round presence of actively feeding planktivores [52,57], and changes to the coastal system can have compounding affects to the rural community. Further monitoring and education of HABs is needed in the region to ensure ecosystem, human, and wildlife health.

## 4. Methods

### 4.1. Study Site

The coastal waters of Southern Mozambique feed the Agulhas Current, one of the strongest WBSs in the world [106,107,108]. This current system is influenced by the East Madagascar Current and eddies within the Mozambique Channel [108]. Water flowing south within the Mozambique Channel along the steep slope and narrow shelf [109] forms both cyclonic and anticyclonic eddies [108]. Westward cyclonic (cold core) eddies are common along the south of Madagascar during Austral winter with a lifetime of a weeks to months [60,66]. These waters are primarily oligotrophic; however, mesoscale features such as coastal upwelling caused by wind stress and cyclonic, cold-core eddies promote increased primary production along the southeastern coast of Africa [61,107].

The study site (Figure 1) was located along the coast of Inhambane Province, Mozambique, stretching approximately 40 km from Barra Beach (23°47′35.00 S, 35°31′05.00 E) to Praia de Jangamo (24°03′52 S, 35°29′33 E). The region is defined by a narrow continental shelf with a steep slope [109], warm waters (21–29 °C), and wind-driven coastal upwelling [107], providing for a diverse coral reef complex. This semi-tropical system is located slightly south of the Tropic of Capricorn and exhibits many characteristics of tropical systems, including abundant light availability and the occurrence of tropical species such as nitrogen-fixing *Trichodesmium* [110] and hard corals [111]; however, the system experiences annual fluctuations in sea surface temperature similar to temperate areas. Based on the geophysical and benthic characteristics, four regions were chosen for detailed study: northern reefs, central reefs, soft substrate, and southern reefs. The northern reefs are dominated by hard plated corals such as *Acropora* spp. (plate coral), *Porites* spp. (porous coral), and *Favia* spp. (false honeycomb coral). The site is marked by strong currents typically flowing from the northwest to the southeast and is also the closest location to Baia de Inhambane and the estuary system. Reefs within the north site range between depths of 23 and 31 m. While the area provides habitat for large aggregations of schooling fish, due to the variable currents, this region is less frequented by recreational divers.

The central reef sites are sheltered within Praia do Tofo coastal bay. Moderate currents flow from north to south along the coast, allowing for the development of branching hard corals such as *Tubastrea micranthus* (green tree coral), as well as soft corals such as *Dendronephthya* spp. (branching soft coral). Depth at reefs ranged between 18 and 30 m. Based on its close proximity to Praia do Tofo and depth, this location is heavily trafficked by recreational divers and fishermen.

The central soft substrate region ranging from 14 m to 18 m depth is located on the leeward side of Tofino Point, approximately 5 km south of Praia do Tofo. The site is an established year-round feeding location of *R. typus*, *M. birostris*, and *M. alfredi,* and it is frequently visited by recreational snorkelers.

The southern reefs are dominated by rocky substrate and soft corals, such as *Lobophytum* spp. (leather coral), *Sarcophyton* spp. (fleshy soft coral), and *Dendronephthya* spp. (branching soft coral). The reefs are located approximately 20 km south of Praia do Tofo and at a depth of 18–27 m.

### 4.2. Satellite Imagery and Analysis

MODIS-Aqua satellite Level 2 images were obtained from the NASA Goddard Space Flight Center (GSFC) Ocean Color data (https://oceancolor.gsfc.nasa.gov, accessed on the 26 September 2018). All images within the sampling period were visually assessed for cloud contamination. Data were processed using SeaDAS v7.4 and re-projected with a 1 km spatial resolution.

### 4.3. Upwelling Index Calculations

Upwelling index (UI), characterized by the volume of water transported through an along-shore transect by a time and distance unit, was calculated using the methodology proposed by Bakun [112] as indicated in technical reports published by the Instituto Español de Oceanografía [113,114]. Sea level pressure field over the ocean is required to calculate UI; however, this study used meteorological data from four coastal World Meteorological Organization (WMO) stations (Vilanculos, #67315; Inhambane, #67323; Panda, #67327; and Xai-Xai, #67335) to calculate wind force components, which were subsequently used to approximate UI at −23.5 °S, 35.3 °E. Pressure data used for the UI calculation were daily average values (hourly). The meridional (*v*) and zonal (*u*) wind components were calculated using the following equations:(1)$$v=\frac{R\cdot T}{p_{a}}\cdot \frac{\Delta p_{x}}{L\cdot f}$$
(2)$$u=\frac{R\cdot T}{p_{a}}\cdot \frac{\Delta p_{y}}{L\cdot f}$$
where R is the ideal gas constant (*R* = 287 j Kg^−1^ K^−1^), *T* is the standard atmosphere temperature (*T* = 288 K), *p*_a_ is standard atmospheric pressure (*p*_a_ = 1013.5 hPa), D*p*_x_ and D*p*_y_ are, respectively, the zonal and meridional pressure difference (hPa), *L* (km) is the distance between the locations used to calculate pressure gradient, and f is the Coriolis parameter defined as twice the vertical component of the Earth’s angular velocity (Ω = 7.3×10^−5^ rad s^−1^) about the local vertical or *f* = 2Ωsin (*θ*) at latitude *θ*.

Ekman transport (*Q,* m^3^ s^−1^ km^−1^) was calculated using the components of wind speed (*u* and *v*), seawater density (*ρ*_w_ = 1025 kg m^−3^), a dimensionless empirical drag coefficient (*C*_d_ = 1.4 × 10^−3^), and air density (*ρ*_a_ = 1.2 kg m^−3^, normal conditions) by means of
(3)$$Q_{x}=\frac{\tau_{y}}{f\cdot \rho_{w}}$$
(4)$$\tau_{y}=\rho_{a}\cdot C_{d}\cdot \sqrt{u^{2}+v^{2}}\cdot v$$
where the *x* subscript corresponds to the zonal component and the y subscript to the meridional component. UI is defined as –*Q*_x_ (m^3^ s^−1^ km^−1^), which is the volume transported per distance unit of an alongshore section. The sign of Ekman transport is changed to define positive (negative) values of UI as response of upwelling (downwelling) favorable winds.

### 4.4. Field Site Monitoring and Sample Collection

Plankton surveys along the coast of Praia do Tofo were designed to investigate the temporal and spatial patterns of DA-producing *Pseudo-nitzschia* abundance. Ships of opportunity, through Peri-Peri Divers, were used to conduct environmental and plankton sampling between January 2017 and August 2018. A pilot monitoring period consisted of collection of whole water samples at least monthly in the soft substrate region pDA analyses from January 2017 to April 2018. Sea surface temperatures were also measured at this time. Measurements and sample collections were a part of a citizen scientist initiative, in conjunction with All Out Africa and Peri-Peri Divers, to explore the plankton community in a known *R. typus* feeding location. An intense study took place from the 20 May 2018 to 10 August 2018. During this time, sampling was planned in each region approximately once per week within an 11 week period; however, weather and access limited sampling events. There was a total of nine sites within the four regions. Samples were collected as the boat drifted above the reef site. Physicochemical parameters (T, conductivity, and DO) were recorded at GPS waypoint markings at the beginning of each sample collection using a YSI ProDSS multiparameter water quality meter (model #626870-1, #627150-4). Calibrations for conductivity (using certified calibration solution at 50,000 µS/cm) and DO were completed prior to each use. Weather activity, wind direction, percent cloud cover, and surface activity (Beaufort scale) were also recorded. Wind data were gathered from World Meteorological Organization station #67323 located at the Inhambane Airport (FQIN).

Approximately 30 L of water was collected at a depth of 5 m via Niskin (General Oceanics Model 1010) during each sampling event. Roughly 1 L of whole water was transferred to an acid-washed container for nutrient analysis. Samples for dissolved inorganic N (NO_2_, NO_3_, and NH_4_), P (PO_4_), Si, and iron (Fe) analysis were filtered through glass fiber filters using gentle vacuum and collected in water quality sterile sample bags (VWR # 89085-546) and frozen. The remaining water was gently poured through consecutive 80 µm mesh (i.e., large fraction) and 36 µm mesh (i.e., small fraction). The larger 80 µm size mesh was used to separate zooplankton from *Pseudo-nitzschia* spp. cells based on the assumption that *Pseudo-nitzschia* cells would pass through, which was confirmed by microscopic analysis. The smaller 36 µm mesh was used to target *Pseudo-nitzschia* cells for pDA analysis. The particulate matter collected on the 80 µm and 36 µm mesh (hereafter referred to as “large” and “small” fraction, respectively) was transported on ice and in the dark to All Out Africa Marine Research Station for processing and preservation. Subsamples of small (duplicate) and large (single) fractions for pDA analysis were filtered onto 25 mm glass fiber filters and frozen. A subsample of the small fraction was preserved with acidified Lugols and used to determine *Pseudo-nitzschia* spp. abundance using the Utermöhl method [115]. Duplicate subsamples from both small and large fractions were used for organic C and N analysis; samples were filtered through combusted glass fiber filters under gentle vacuum and frozen for later analysis. Analyses were conducted following standard procedures by Bianchi and Bauer [116]. All samples were frozen or preserved and transported to Louisiana State University or University of Minnesota Duluth Large Lakes Observatory for analysis.

### 4.5. Dissolved Nutrient Analysis

Ammonium, nitrate + nitrite, and orthophosphate concentrations were determined using standard AutoAnalyzer techniques (EPA 353.4, EPA 350.1, and EPA 365.5). Dissolved inorganic N was calculated as the sum of nitrate, nitrite, and ammonium concentrations, and dissolved inorganic P was considered equal to the orthophosphate concentration. Fe and Si were analyzed by ICP-OES using standard methods (EPA #200.7). Quantification limits were 0.1 µM for nitrate, 0.2 µM for nitrite, 0.1 µM for ammonia, 0.04 µM for P, 0.012 mg L^−1^ for Si, and 0.007 mg L^−1^ for Fe.

### 4.6. Pigment Analysis

Duplicate samples for chla were filtered through glass fiber filters under gentle vacuum for the small fraction. Filters were frozen and stored for photopigment analysis by high performance liquid chromatography (HPLC) following the method from Pinckney et al. [64]. Filters were sonicated in 3 mL of 100% acetone for 30 s and extracted in the dark for 20 to 24 h at −20 °C. Extracts were filtered through 0.2 µm filter, and 300 µL was injected into an HPLC system equipped with reverse-phase C18 columns in series (Rainin Microsorb-MV, 0.46 × 10 cm, 3 mm, Vydac 201TP, 0.46 × 25 cm, 5 mm). A nonlinear binary gradient adapted from Van Heukelem et al. [117] was used for pigment separations. Solvent A consisted of 80% methanol and 20% ammonium acetate (0.5 m adjusted to pH 7.2), and solvent B consisted of 80% methanol and 20% acetone. Absorption spectra and chromatograms were acquired using a Shimadzu SPD-M10av photodiode array detector, where pigment peaks were quantified at 440 nm.

Photopigment concentrations were examined with CHEMTAX (v.1.95) [65] to determine the absolute abundance of major phytoplankton groups (µg chl *a*^−1^) using the convergence procedure outlined by Latasa [118]. The initial pigment matrix was derived from Barlow et al. [60] and modified using Higgins et al. [119] (Appendix A). Measured pigments include chlorophyll *a* (chla), chlorophyll *b* (Chl *b*), chlorophyll *c*_1_ (Chlc1), chlorophyll *c*_2_ (Chlc2), chlorophyll *c*_3_ (Chlc3), peridinin (Per), 19′-butanoyloxyfucoxanthin (But), fucoxanthin (Fuc), neoxanthin (Neo), violaxanthin (Viol), prasinoxanthin (Pras), 19′-hexanoylfucoxanthin (Hex), alloxanthin (Allo), zeaxanthin (Zea), antheraxanthin (Anth), and lutein (Lut). Previous research suggests that the majority of the phytoplankton community is represented within the 20–200 µm size range [120]; phytoplankton groups whose size were <36 µm were excluded from the matrix. Pigment ratio values for cyanobacteria were derived from Higgins et al. [119] and reported as *Trichodesmium*. *Pseudo-nitzschia* spp. was distinguished from other diatoms through the presence of Chlc1 [119].

A portion of the small fraction samples (13%) were analyzed via microscopy to verify CHEMTAX results. A settling procedure [121] was used to concentrate the samples to 5 mL, and cell counts were performed using a Sedgwick–Rafter counting cell on an inverted microscope at 10× objective (ZIESS Axis Observer-A1 axiovert 135) and identified to the genus level. A subsample of the large fraction was collected and preserved with 5% acidified Lugols, and organisms were identified to the genus level using a dissecting scope (ZIESS SteREO Discovery.V8). *Chaetocerous* spp. were noted within several samples among the large fraction; however, due to preservation effects, biomass could not be accurately quantified, so presence/absence was noted.

### 4.7. Particulate Domoic Acid Analysis

Concentration of pDA was assessed within three months of collection by Biosense ASP ELISA (Amnesic Shellfish Poison, enzyme-linked immunosorbent assay) (Biosense #A31300401) using the protocol described in Baustian et al. [122] expressed as pg pDA L^−1^. Samples were extracted using 20% methanol, sonicated, then filtered to remove cell debris. Plates were read using a Thermo Varioskan Flash spectral scanning multimode reader and SkanIt Software version 2.4.1. Toxin concentrations (pDA) were calculated using a non-linear 4-parameter logistic curve fit provided by Abraxis (Abraxis, Inc, Warminster, PA, USA). Detection limit was 9.33 ± 1.1 pg DA mL^−1^ across the analyzed plates; concentrations below detection limit were reported as below detection limit.

### 4.8. Statistical Analysis

Multivariate analyses were performed using SigmaPlot v14 (Inpixon, Palo Alto, CA, USA). Distinct one-way analysis of variance (ANOVA) for temperature, dissolved oxygen, and salinity parameters were used to compare regions. Pearson correlations (Appendix A) for physicochemical and biological variables in surface water throughout the study were tested. Significance was defined as a *p* value < 0.05. PCA analyses were performed to characterize the relationships between physicochemical parameters (T, salinity, and nutrients) and biological parameters (chla, *Pseudo-nitzschia* cell counts, and pDA).

## Figures and Tables

**Figure 1 toxins-13-00903-f001:**
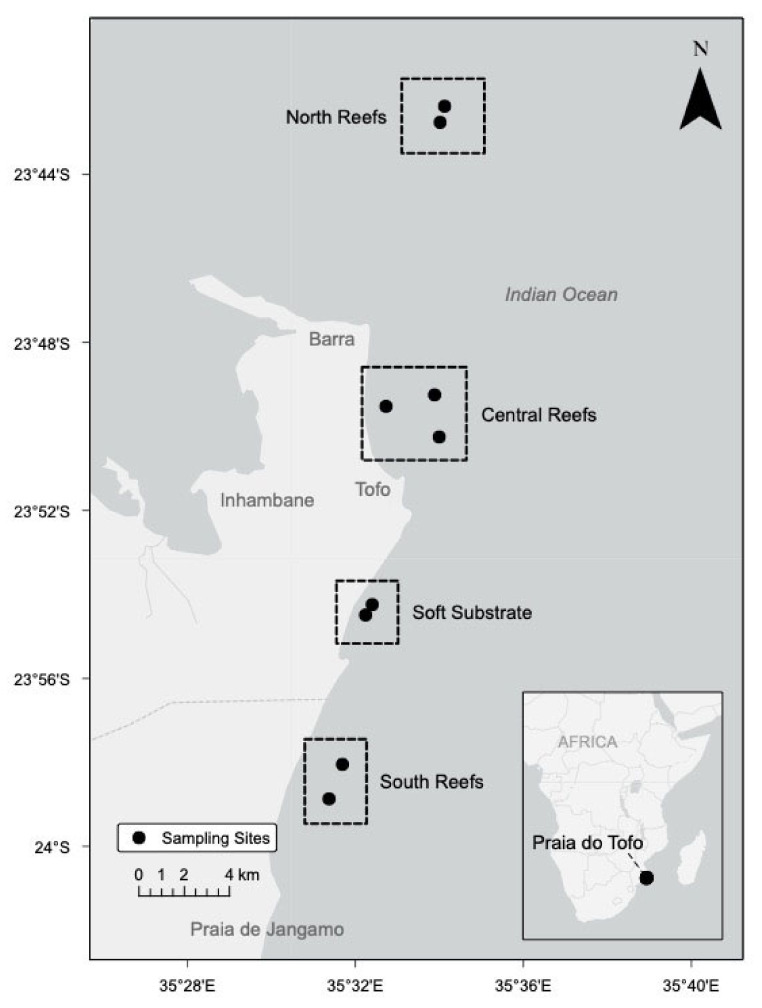
Study site: Inhambane Province, Mozambique. Four areas within the region were identified between Barra and Praia de Jangamo (40 km). During the pilot monitoring period between January 2017 and April 2018, only the soft substrate region was sampled. During the intensive study period, all four regions were sampled, but no statistical significance was noted.

**Figure 2 toxins-13-00903-f002:**
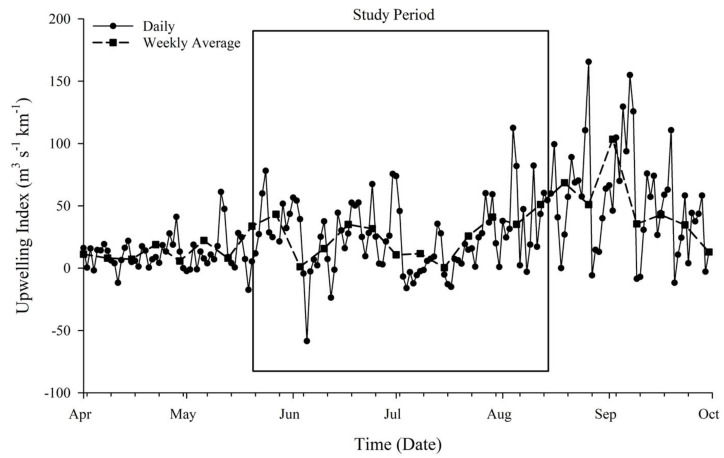
Calculated upwelling index. Atmospheric pressure data were used to estimate upwelling transport flux (m^3^ s^−1^ km^−1^) along the coast of Inhambane Province. The dotted line shows the weekly average of transport flux. The box represents the timeframe of the intensive study period. A prolonged increase in upwelling flux occurred in late July through August, corresponding to multiple biological responses, referred to as a major upwelling event (MUE).

**Figure 3 toxins-13-00903-f003:**
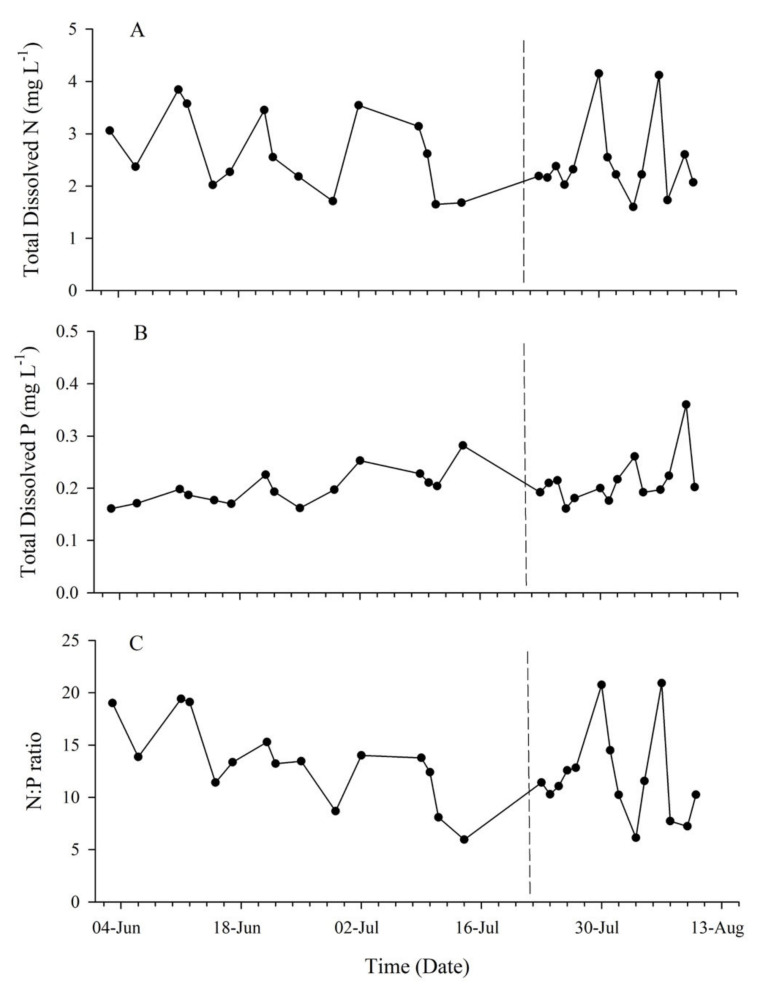
Inorganic dissolved nutrients. (**A**) Total dissolved nitrogen (N), (**B**) total dissolved phosphorus (P), and (**C**) N:P ratio over the intense study period (May–August 2018) indicated the system is primarily controlled by nitrogen limitation. The dotted line indicates the MUE that occurred following 19 July 2018.

**Figure 4 toxins-13-00903-f004:**
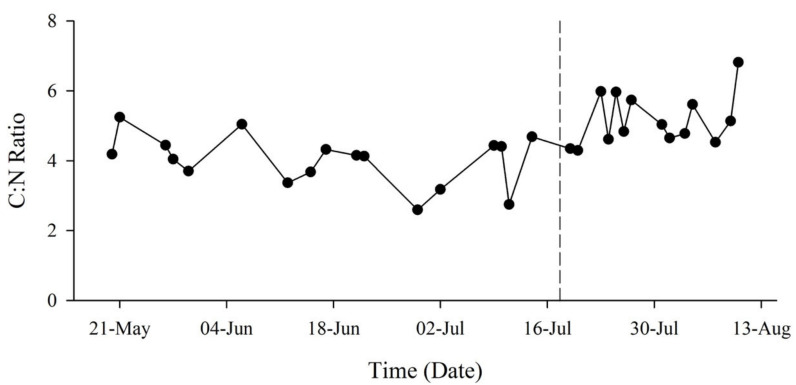
Particulate carbon (C) to N ratio (C:N) from May to August 2018. Total (combined fractions) ratios fluctuated throughout the study period, but average ratio increased following the MUE (dotted line).

**Figure 5 toxins-13-00903-f005:**
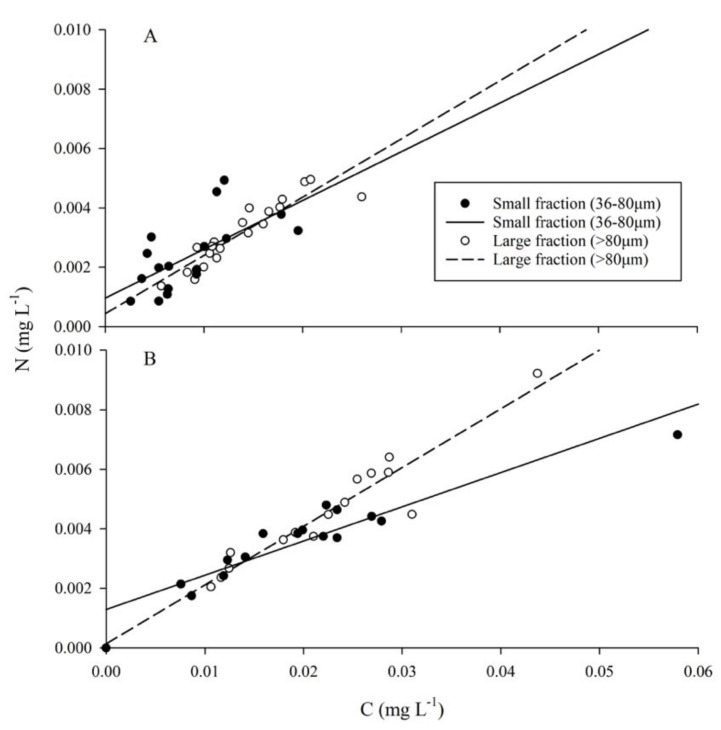
Particulate organic matter (POM)**.** (**A**) Prior to the MUE, regressions between cellular C:N had coefficients of 0.144 (*R*^2^ = 0.395) and 0.196 (*R*^2^ = 0.836) for small and large fractions, respectively. (**B**) Following the MUE, the small fraction showed a decreased coefficient of 0.102 (*R*^2^ = 0.881), whereas the large fraction maintained a similar relationship of 0.197 (*R*^2^ = 0.90).

**Figure 6 toxins-13-00903-f006:**
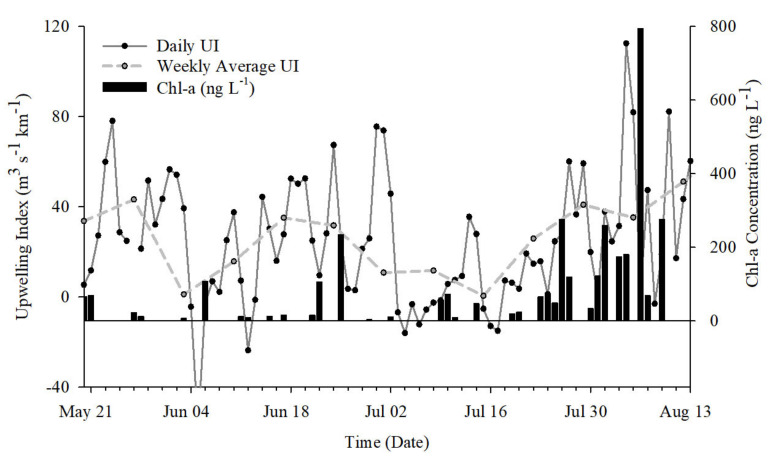
Chlorophyll *a* (chla) concentrations and upwelling index for May to August 2018. Chla, used as a proxy for primary productivity, increased in late July into August, corresponding to MUE.

**Figure 7 toxins-13-00903-f007:**
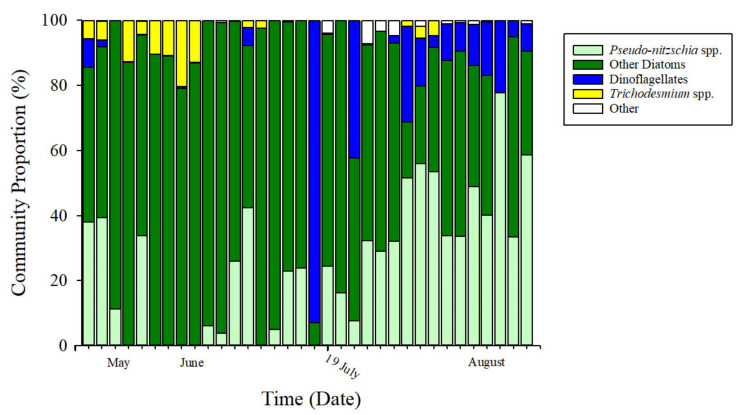
Phytoplankton assemblage composition. CHEMTAX analysis [65] revealed the assemblage was diatom-dominated (shades of green) throughout the study period. Following the MUE on 19 July 2018 (indicated on timeline), there was an increase in assemblage dominance by *Pseudo-nitzschia* spp. (light green).

**Figure 8 toxins-13-00903-f008:**
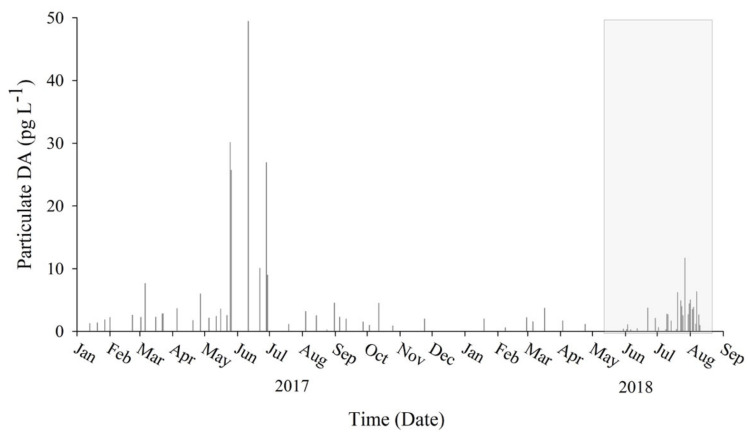
Particulate domoic acid (pDA) concentration in coastal Inhambane Province, Mozambique. Samples collected from January 2017 through August 2018 detected toxin year-round. The shaded box highlights samples collected during the intensive study during May–August 2018. Data suggest a possible spike in pDA production during the onset of Austral winter.

## Data Availability

Role of Coastal Environmental Conditions During Austral Winter on Plankton Community Dynamics and the Occurrence of *Pseudo-nitzschia* spp. and Domoic Acid in Inhambane Province, Mozambique. Available online: https://digitalcommons.lsu.edu/gradschool_theses/5056 (accessed on 15 November 2021).

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
