# Peer review of "Domoic Acid and Pseudo-nitzschia spp. Connected to Coastal Upwelling along Coastal Inhambane Province, Mozambique: A New Area of Concern"

_toxins, 2021, doi:10.3390/toxins13120903_

Round 1

Reviewer 1 Report

The manuscript “Domoic acid and Pseudo-nitzschia spp. connected to coastal upwelling along coastal Inhambane Province, Mozambique: a new area of concern” is a comprehensive study on toxic microalgae in the coast of Mozambique and identifies for the first time the neurotoxin domoic acid in this area. The research describes the main factors involved in phytoplankton assemblages and abundances as well as their link to upwelling system and productivity. Although this is an important fishing ground and a marine biodiversity hotspot the coasts of Mozambique are still quite understudied and therefore, this article provides new and valuable data for science and stakeholders. Furthermore, the manuscript is very well written and the methodology and statistical analyses accurately presented. I would accept this article in its present form except for a few minor remarks, which may be corrected during proof editing stage. These are:

Line 7-9: “no incidences of marine phycotoxins have been recorded in Mozambique, which may be due to absence of a monitoring program…”     It seems contradictory. Monitoring programs aim at avoiding incidents, in fact.

Line 222-223 “when N is high, and N:P ratio is low, pDA is higher” It seems ambiguous. Correct or explain further.

Check capitals in references.

Author Response

Line 7-9: changed "incidences" to "occurrences" 

Line 222: reworded sentence

References checked and corrected capitalizations and a few spelling mistakes.

Reviewer 2 Report

Manuscript entitled „ Domoic acid and Pseudo-nitzschia spp. connected to coastal  upwelling along coastal Inhambane Province, Mozambique: a  new area of concern” is an interesting, well-written and well-planned experimental work. However, the text needs many corrections according to the following comments:

Introduction

line 43 – use abbreviation DA instead of full name domoic acid

line 49 – remove space after “programs”

line 53 – write Pseudo-nitzschia in italics

line 57 - remove space after dot and before While

line 58 – use an increase instead of increases

line 73 – is Pseudo-nitzschia multiseries is plural should be “have been” not has been

line 85 – why is SCUBA written in capital letters?

line 92 - explain in full name abbreviations N, P, Si

Results

line 104 – write (T; p = 0.81) and (DO; p = 0.76)

line 119 – write 14th of July 2018

line 120 – write 14th of July

line 128 – use only abbreviation T and DO because their full names were explained earlier in the text of manuscript

line 130 - enter the values in which the salinity was calculated

line 130 - write 14th of July 2018

line 132 – put spaces before and after ±

line 134 – use abbreviation N instead of nitrogen

line 135 – explain abbreviations in full name, use abbreviation N instead of nitrogen

line 136 – remove space between 3 and A

line 139 - use only abbreviation P

line 140 - remove space between 3 and B

line 145 - use abbreviation N instead of nitrogen, remove space between 3 and C

line 152 – explain abbreviation C in full name

line 154 – write abbreviations C and N instead their full names

line 156 - remove space between 5 and A

line 167 – what abbreviation POM means, please write the full name

line 167, 173 - remove space between 5 and B

line 174 - write 6th of August 2018

line 180 – decide which type of abbreviation for chlorophyll a you will use (chla) or (Chl-a) like in figure 6

line 181 – what means SST – explain this abbreviation

line 191 – write were instead was

line 209 – this sentence “These preliminary data 209 suggest a possible seasonal spike at the onset of the austral winter months with peaks in 210 June of 2017 and July of 2018” should be located in the Discussion section

line 213 - write 11th of June 2017

line 218 – write 27th of July 2018

line 219, line 221 - put spaces before and after ±

Figure 4 – write a full name of abbreviation C

Figure 6 – write Concentrations and Upwelling with a lowercase letter

Discussion

line 230, 263, 265, 290, 291, 295 – if you decide to use abbreviation N for nitrogen you should use this abbreviation later in the whole text

line 234 - – if you decide to use abbreviation DA for domoic acid you should use this abbreviation later in the whole text

line 235 – remove and after event and put it between SST and pDA

line 261 – remove one as from sentence, because it is doubled

line 271 - remove space between 3 and C

line272 – put it before was

line  273 - remove space between 5 and B

line 290 - put spaces before and after ±

line 291 - if you decide to use abbreviation P for phosphorus you should use this abbreviation later in the whole text; put spaces before and after ±

line 292 – use abbreviation PCA instead of Principle component analysis

line 297 – write revealed

line 315 – put each on the beginning of sentence

line 323 – add region after soft substrate

Materials and Methods

taking into account the previous comments on how to write the date and the use of abbreviations according to the rules of the journal “Acronyms/Abbreviations/Initialisms should be defined the first time they appear in each of three sections: the abstract; the main text; the first figure or table. When defined for the first time, the acronym/abbreviation/initialism should be added in parentheses after the written-out form”, please correct the text of this part of the manuscript in line 448, 454, 462, 462, 468, 476, 486, 484, 485, 486, 488, 490, 506, 514, 532, 539

References

for example line 547, 549, 553, 555, and almost all positions - write abbreviated journal name with dots, please prepare a literature list in accordance with the rules set out by the journal

line 590 – in position 21 put the name of journal, issue and pages

line 634 – in position 40 put the number of pages

line 668, 673, 766, 782 - in position 54, 56 put the number of pages

Author Response

Thank you for your detailed and thorough comments!

Round 2

Reviewer 2 Report

I recommend the publication of article